# Effects of Macronutrients on the In Vitro Production of ClpB, a Bacterial Mimetic Protein of α-MSH and Its Possible Role in Satiety Signaling

**DOI:** 10.3390/nu11092115

**Published:** 2019-09-05

**Authors:** Manon Dominique, Jonathan Breton, Charlène Guérin, Christine Bole-Feysot, Grégory Lambert, Pierre Déchelotte, Sergueï Fetissov

**Affiliations:** 1TargEDys SA, Faculty of Medicine and Pharmacy, University of Rouen Normandy, 22, Boulevard Gambetta, Cedex 01, 76183 Rouen, France (M.D.) (G.L.) (P.D.); 2Nutrition, Gut and Brain Laboratory, Inserm UMR1073, University of Rouen Normandy, 76183 Rouen, France (J.B.) (C.G.) (C.B.-F.); 3Institute for Research and Innovation in Biomedicine (IRIB), University of Rouen Normandy, 76183 Rouen, France; 4Rouen University Hospital, CHU Charles Nicolle, 76183 Rouen, France; 5Laboratory of Neuronal and Neuroendocrine Differentiation and Communication, Inserm UMR1239, University of Rouen Normandy, 76130 Mont-Saint-Aignan, France

**Keywords:** gut microbiota, satiety hormones, macronutrients, *E. coli*, ClpB, PYY

## Abstract

Gut microbiota can influence the feeding behavior of the host, but the underlying mechanisms are unknown. Recently, caseinolytic protease B (ClpB), a disaggregation chaperon protein of *Escherichia coli*, was identified as a conformational mimetic of α-melanocyte-stimulating hormone (α-MSH), an anorexigenic neuropeptide. Importantly, ClpB was necessary for *E. coli* to have an anorexigenic effect in mice, suggesting that it may participate in satiety signaling. To explore this further, we determined the short-term (2 h) effects of three macronutrients: protein (bovine serum albumin), carbohydrate (D-fructose) and fat (oleic acid), on the production of ClpB by *E. coli* and analyzed whether ClpB can stimulate the secretion of the intestinal satiety hormone, peptide YY (PYY). Isocaloric amounts of all three macronutrients added to a continuous culture of *E. coli* increased ClpB immunoreactivity. However, to increase the levels of ClpB mRNA and ClpB protein in bacteria and supernatants, supplementation with protein was required. A nanomolar concentration of recombinant *E. coli* ClpB dose-dependently stimulated PYY secretion from the primary cell cultures of rat intestinal mucosa. Total proteins extracted from *E. coli* but not from ClpB-deficient *E. coli* strains also tended to increase PYY secretion. These data support a possible link between *E. coli* ClpB and protein-induced satiety signaling in the gut.

## 1. Introduction

The composition of the gut microbiota has been associated with host metabolic phenotypes. For instance, colonization of germ-free mice with gut microbiota harvested from conventionally raised obese or lean mice results in similar host fatness to that of the donor mice [1]. The underlying mechanisms are not completely understood. They may include both direct energy-extracting properties of the gut bacteria as well as the regulation of energy metabolism by bacteria-host interactions, including host appetite and nutrient intake [2]. For instance, proteins produced by commensal bacteria, such as *Escherichia coli*, were shown to stimulate satiety signaling in rodents [3]. The increased prevalence of *E. coli* in anorexic compared to obese individuals also supports a possible role of these common gut bacteria in the promotion of host satiety signaling [4]. 

The intestinal satiety pathways include the vagal afferents and the enteroendocrine cells located throughout the intestinal epithelium that secrete a variety of satiety hormones including peptide tyrosine–tyrosine (PYY). It is currently accepted that meal-derived macronutrients, i.e., protein, fat and carbohydrates, as well as products of their digestion, activate the intestinal satiety pathways by binding to receptors expressed by the enteroendocrine cells [5]. Of all the macronutrients, proteins have the best effect on satiety and are known to preferentially activate enteroendocrine L-cells to secrete PYY [6].

Gut bacteria, such as *E. coli,* reside on the surface of the gut epithelium suggesting that some bacterial metabolites and bacterial compounds generated after natural bacterial lysis can directly activate intestinal satiety pathways. It also suggests that gut bacteria could play a role in the interface between the nutrients and intestinal satiety pathways. Bacteria typically respond to nutrients by dividing and growing exponentially until they reach a threshold density of about 10^9^–10^12^ cells/mL. At this point they enter the stationary growth phase. We have recently shown that supplying *E. coli* with regular nutrients, imitating two daily meals, induces immediate growth that continues for 20 min before entering the stationary phase. A similar dynamic of bacterial growth was observed after infusion of a nutritional medium in the rat colon, and moreover, infusion of *E. coli* proteins 2 h after onset of the stationary phase stimulated PYY secretion [3]. These data show that nutrient-induced bacterial growth dynamics in the gut correspond temporally to the activation of the intestinal satiety pathways, including PYY secretion. This suggests a functional role of gut bacteria in the regulation of host appetite [2]. 

Some metabolites such as short chain fatty acids (SCFA) produced during gut bacterial fermentation of mainly non-digestible fibers have been shown to activate PYY secretion [7]. It is, however, unknown whether *E. coli* can participate in macronutrient-induced activation of PYY, i.e., in triggering one of the main humoral satiety pathways. Our interest in the possible link between *E. coli* and host satiety began after the *E. coli* caseinolytic protease B (ClpB) protein was identified as a conformational mimetic of α-melanocyte-stimulating hormone (α-MSH), an anorexigenic neuropeptide [8]. ClpB is a 96 KDa heat shock chaperon that protects bacteria from protein aggregation [9]. The ClpB protein sequence has an α-MSH-like motif recognized by the α-MSH antibody. Importantly, ClpB was necessary for an anorexigenic effect and the reduction of body weight seen in mice treated intragastrically with *E. coli* [8]. This suggests that ClpB may participate in *E. coli*-mediated intestinal satiety signaling, but questions remain as to whether ClpB production in these bacteria is dependent on the macronutrient composition and whether ClpB can stimulate intestinal satiety pathways.

Thus, to address, at least partly, these questions, in the present work, we studied a possible role of *E. coli* ClpB in mediating the macronutrient led PYY secretion. For this purpose, we first analyzed the short-term in vitro effects of the three macronutrients (protein, carbohydrate and fat) on the production of ClpB by *E. coli*. Then we determined whether ClpB could directly stimulate the primary cell culture of rat intestinal mucosa to secrete PYY.

## 2. Materials and Methods

### 2.1. E. coli Bacterial Culture and Macronutrient Supplementation

*E. coli* K12 bacteria were cultured for 48 h at 37 °C in 40 mL of Mueller-Hinton (MH) medium (Sigma-Aldrich, St. Louis, MO, USA) composed of beef infusion solids (2.0 g/L), casein hydrolysate (17.5 g/L), starch (1.5 g/L) at pH 7.4. Every 12 h, after a centrifugation step (4300 rpm, 5 min), the supernatant was discarded, and bacteria were supplemented by an equivalent volume of MH medium imitating two daily meals (Figure 1). Bacterial growth rate was monitored every hour by measuring the optical density using a spectrophotometer (BioMate, ThermoElectron Corporation, Waltham, MA, USA).

After 48 h of incubation (t0h), 20 mL of bacterial culture were taken and used for protein (10 mL) and the RNA (10 mL) extraction from the bacterial pellets after centrifugation (4300 rpm, 30 min, 4 °C). The remaining 20 mL were centrifuged (4300 rpm, 5 min), the supernatant was discarded and 20 mL of fresh MH medium or isocaloric amounts of macronutrients were added to the bacterial culture. This isocaloric amount was fixed at 1.68 calories, corresponding to the caloric value of 20 mL of MH medium.

All four experimental conditions were compared in quadruplicate: (1) “Control”, 20 mL of MH medium; (2) “Protein”, bovine serum albumin (BSA, Sigma-Aldrich, Mo, USA; 4 cal/g) was diluted in water at a concentration of 1.32 g/mL and 318 µL was adjusted by water to 20 mL; (3) “Carbohydrate”, D-fructose (Sigma; 3.73 cal/g) was diluted in water at a concentration of 1.69 g/mL and 266 µL was adjusted by water to 20 mL; (4) “Fat”, oleic acid (Sigma; 9.46 cal/g) was diluted in water at a concentration of 0.887 g/mL and 203 µL were adjusted by water to 20 mL. Although the food-derived oleic acid is mainly present in form of esters, the direct use of oleic acid was justified by the data showing that gut appetite-regulatory effects of lipids can be mediated by free fatty acids produced after hydrolysis of triglycerides [10,11]. 

After adding of macronutrients, the samples were incubated by stirring at 37 °C for 2 h. After 2 h of incubation (t2h), the cultures were divided in half (10 mL for protein and 10 mL for RNA extraction). After centrifugation (4300 rpm, 30 min, 4 °C), 1 mL of supernatant was recovered for the ClpB assay while protein and RNA were extracted from bacterial pellets as described below.

### 2.2. Immunocytochemistry

To visualize ClpB presence in bacterial cells we have used immunocytochemistry. A 20 µL sample from the bacterial pellet of each experimental condition (*n* = 4) at t0h and t2h (MH, BSA, D-fructose, oleic acid) was plated on glass slides and the bacteria were fixed with 100% ethanol at 37 °C. The glass slides were washed 3× with 0.3% phosphate-buffered saline (PBS)/Triton solution, and preincubated for 1 h with a blocking solution (PBS/5% BSA/0.3% Triton/0.2% NaN_3_). For detection of ClpB, slides were incubated with anti-ClpB monoclonal antibody (Delphi Genetics, Gosselies, Belgium) (1:50) for 1.5 h and after three washes with 0.3% PBS/Triton solution, with a secondary anti-mouse antibody (1:200) conjugated to fluorescein isothiocyanate (FITC) (Jackson ImmunoResearch, Ely, Cambridgeshire, UK). Two drops of 4′,6-diamidino-2-phenylindole (DAPI) (Sigma) were added to each slide to counterstain the bacterial DNA. The ClpB-immunofluorescence and DAPI staining were observed under a fluorescence microscope (Zeiss, Oberkochen, Germany) and images were taken using ×100 objective from five areas/slide. The presence of ClpB was analyzed by quantifying the relative number of ClpB positive bacteria on the total number of DAPI stained bacteria on the visual field using ImageJ^®^ software (LOCI, University of Wisconsin-Madison).

### 2.3. ClpB Concentration

The bacterial pellet was dissolved in a protein extraction buffer (PBS + 1% of a protease inhibitor cocktail, Sigma), sonicated (20% amplitude for 30 s) and centrifuged (12.000× *g*, 5 min, 4 °C) to separate cellular debris from the cytoplasmic content. The latter was stored at −80 °C until the ClpB assay. ClpB protein concentrations were measured by the enzyme linked immunosorbent assay (ELISA) previously validated and described in detail [3]. For this purpose, three antibodies were used: rabbit polyclonal anti-ClpB (Delphi Genetics) as a capture antibody, mouse monoclonal anti-ClpB as a detection antibody (Delphi Genetics) and an alkaline phosphatase-conjugated goat anti-mouse revelation antibody (Jackson ImmunoResearch). The optical density of the ELISA reaction was measured at 405 nm using a microplate reader (Metertech 960, Taipei, Taiwan) and the ClpB concentration was determined using a ClpB (Delphi Genetics) standard curve. Each determination was performed in duplicate.

### 2.4. ClpB mRNA Expression

Total bacterial RNA was extracted in cold TRIZOL reagent (Invitrogen, Carlsbad, CA, USA). After extraction, RNA concentrations were measured using a NanoDrop spectrophotometer (ThermoScientific, Waltham, MA, USA). To generate cDNA, a reverse transcription reaction was performed with 1 µg of total RNA using of M-MLV reverse transcriptase (200 U/µL) (ThermoFisher). Quantitative polymerase chain reaction (PCR) was performed on all samples using a BioRad CFX96 Real Time PCR System (BioRad, Hercules, CA, USA) and SYBR Green Master Mix (Life Technologies, CA, USA). The primers for *ClpB* cDNA amplification were: 5′-GGAAAAGCAACTGGAAGCCG-3′ and 3′-TACGCGATGGTGCAGTTCTT-5′ (Tm = 60 °C) as previously described [8]. The relative, between groups, levels of ClpB mRNA expression were estimated by the inverse values of the amplification cycle threshold (Ct) for each cDNA sample curve.

### 2.5. Rat Primary Intestinal Culture

Animal experimental procedures were approved by the Local Ethical Committee of Normandy (approval n°5986). Seven-week old male Sprague Dawley rats were purchased from Janvier Labs (Genest-Saint-Isle, France) and were kept for 1 week in standard rat holding cages (two rats per cage) in a specialized animal facility to acclimatize them to the environmental conditions: 22 °C ± 3 °C, relative humidity 40 ± 20% and a 12/12 h dark/light cycle. All rats were given ad libitum access to drinking water and food (Kliba Nafag, Rheinfelden, Germany).

The protocol of the primary intestinal culture was adapted from Psichas et al. [7]. Rats were killed by decapitation, the colons and ileums were dissected and washed with an ice-cold PBS. Then, the intestinal tissue was cleaned with ice-cold L-15 (Leibowitz medium, Sigma) and digested with 0.4 mg/mL collagenase XI (Sigma) in high-glucose DMEM (+ 1% L-Glutamine + 1% Penicillin + 1% Streptomycin + 1% non-essential amino acids) for 10 min at 37 °C. Cell suspensions were centrifuged (10 min, 400× *g*) and the pellets were resuspended in high-glucose DMEM (the same as previously but with 10% Fetal Bovine Serum (FBS)). Cell suspensions were filtered through a nylon mesh (pore size 100µm, Merck Millipore, Burlington, MA, USA) and plated onto 24-well, 1% Matrigel-coated plates (Corning, NY, US). The plates were incubated overnight at 37 °C in an atmosphere of 95% O_2_ and 5% CO_2_.

### 2.6. Gut Hormone Secretion Experiments

After 24 h of primary culture, intestinal cells were incubated for 20 min in a water bath with a secretion buffer (4 mM KCl, 138 mM NaCl, 1.2 mM NaHCO_3_, 1.2 mM NaH_2_PO_4_, 2.6 mM CaCl_2_, 1.2 mM MgCl_2_ and 10 mM HEPES adjusted to pH 7.4 with NaOH). Recombinant 96 KDa ClpB protein (Delphi Genetics) was added to the secretion buffer at different concentrations in a pico- to a nano-molar range: 1.26 pM, 2.52 pM, 1.26 nM, 13 nM, 25 nM and 130 nM. Cells incubated with PBS were used for control. Additionally, 15 ng/µL of total protein extracted from *E. coli* K12 wild type (WT) or ClpB mutant (ΔClpB) *E. coli* strains, previously generated in the Bernd Bukau’s laboratory (Center for Molecular Biology, Heidelberg University, Germany), were tested in a similar fashion. After incubation for 20 min, the cells were treated with a lysis buffer (50 mmol/L Tris-HCl, 150 mmol/L NaCl, 1% IGEPAL CA-630, 0.5% deoxycholic acid + protease inhibitor cocktail (Sigma). Cells were collected with a cell scraper, the lysates were centrifuged (12.000× *g*, 20 min,) and stored at −80 °C until the PYY assay using a Fluorescent Immunoassay Kit^®^ (Phoenix Pharmaceuticals, Inc, Phoenix, AZ, USA) according to the manufacturer’s instructions.

### 2.7. Statistical Analysis

Statistical analysis was performed using Prism 6.0 software (GraphPad Inc., San Diego, CA, USA). Data were analyzed using two-tailed unpaired *t*-test or Mann–Whitney test and one-way ANOVA (for a comparison between multiple groups) or two-way ANOVA (for a comparison between multiple groups and between t0h and t2h time-points) with Sidak’s multiple comparisons test. Normality was evaluated by the Kolmogorov–Smirnov test. Data are shown as means ± standard error of means (SEM) and *p* < 0.05 was considered statistically significant.

## 3. Results

### 3.1. Macronutrient Effects on ClpB Production and Expression

The bacterial growth dynamics using the 12-hourly MH culture medium provision data are shown in Figure 1. It shows the acceleration of the onset of the stationary phase after the third nutrient provision as previously reported [3]. The fifth nutrient provision, which was previously shown to trigger the onset of stationary phase in 20 min [3], was the starting point for the analysis of the effects of macronutrients on ClpB production (Figure 1).

The immunocytochemical detection of ClpB showed strong immunoreactivity only in a fraction of *E. coli* bacteria (about 14%) at t0h, before nutrient provision (Figure 2A,B). After 2 h of nutrient provision (MH medium or macronutrients), there was a visible increase in the immunoreactivity of ClpB, (Figure 2A,B) reaching about 28% of the total number of bacterial cells. This increase was significant in all experimental groups (Figure 2B). However, no significant differences in the relative increase of ClpB immunoreactivity were seen between the three macronutrient groups (Figure 2C).

The effects of macronutrient provision on ClpB concentration were analyzed in bacterial cells. Whereas, the effects on the secreted fraction of ClpB were analyzed in culture medium supernatants. In bacterial cells, the concentration of ClpB increased at t2h compared to t0h only in the MH and BSA groups (Figure 3A). Moreover, the ClpB concentration in the BSA group was higher than in the control MH and macronutrient groups (Figure 3A).

Thus, the comparison of ClpB concentrations in bacterial cells between t0h and t2h demonstrates significant increases of 192% and 250% in the MH and BSA groups, respectively. No significant changes were found in the D-fructose (+51%) and oleic acid (+65%) groups.

After BSA supplementation, the ClpB protein concentration also increased in the culture medium supernatants compared to the other groups (Figure 3B).

The effect of macronutrients on the expression levels of ClpB mRNA in *E. coli* was analyzed at t0h and t2h after nutrient provision. Supplementation in BSA increased the ClpB mRNA expression (BSA, +18% vs. CTt2h) (Figure 3C). No significant differences in ClpB mRNA levels were observed between these time points among the other groups.

### 3.2. ClpB Effects on PYY Secretion

Incubation of the primary cell culture of rat intestinal mucosa with recombinant *E. coli* ClpB protein showed an increase of PYY secretion in a dose-dependent manner. The PYY secretion was significantly higher from the basal ClpB secretion level (+21%) starting at ClpB concentration of 1.26 nM and it was further increased (+58%) at the ClpB concentration of 130 nM (Figure 4A).

To test the specificity of the ClpB-activating effect on PYY secretion among *E. coli* proteins, a primary cell culture of rat intestinal mucosa was incubated with total proteins extracted from *E. coli* WT or *E. coli* ΔClpB strains. We found that only ClpB-containing *E. coli* protein extracts displayed a tendency to increase PYY secretion (Figure 4B).

## 4. Discussion

The present in vitro study revealed a differential effect of macronutrients on short-term bacterial ClpB production. This was confirmed by showing that protein supplementation was the strongest activator of both ClpB mRNA and protein levels in *E. coli*. Furthermore, we showed that *E. coli* ClpB was able to stimulate PYY secretion in rat intestinal mucosa. Taken together, these data point to the possible role of *E. coli* ClpB in the molecular signaling cascade of protein-induced satiety.

Using immunocytochemistry, we showed that ClpB immunoreactivity was detected only in a fraction of bacteria. This immunoreactivity was dependent on the duration of the stationary phase, indicating that the functional regulation of ClpB during *E. coli*’s nutritional cycles. As such, at the end of the stationary growth phase, which is typically accompanied by the accumulation of various metabolic proteins, there is an increase in protein disaggregation to maintain cellular homeostasis. In other words, it requires increased activity of the chaperone system. To perform protein disaggregation, ClpB is bound to a molecular complex of the DnaK chaperone system [12]. It is likely that such a formation of a macromolecular complex may prevent ClpB detection by its antibody. Thus, although ClpB is expressed in every bacterial cell, its immunodetection may reflect unbound ClpB molecules which are not currently involved in the protein disaggregation process. This may explain a relatively small amount of immunopositive ClpB at t0h which corresponds to the end of the previous feeding cycles and more than 10 h of the stationary phase. In contrast, the beginning of the stationary phase (at t2h) may require less ClpB activity, this would explain its relatively high immunoreactivity. Therefore the immunodetection of ClpB appears to be independent from the ClpB production factor relevant to its functional activity. We did not observe any significant differences between the macronutrients effects on ClpB immunoreactivity. This suggests that the immunoreactivity was mainly linked to the duration of the stationary phase. Nevertheless, a non-significant trend of lower ClpB immunoreactivity in BSA vs. D-fructose or oleic acid groups suggests that protein supplementation may require more ClpB activity, in agreement with its role in protein disaggregation. Most importantly, increased ClpB production at both the mRNA and the protein levels were observed after protein supplementation. The bacteria were fed with a protein-rich, mixed macronutrient source (MH medium). This led to an increase in ClpB production, albeit to a lesser extent than the levels seen after protein only supplementation. These results confirm our previous observations in *E. coli* cultures maintained in a MH culture medium [3] and further reveal a key role of proteins in the stimulation of ClpB production. 

Although the typical food has a mixed macronutrient composition, the relative amount of each macronutrient contributes differently to satiety. As such, a role of protein in the induction of prolonged satiety has been well established and may underlie the efficacy of protein-rich diets aimed at reducing total caloric intake and body weight in overweight and obese subjects [13,14,15,16,17]. The underlying effects may include activation by proteins and their derived peptides and amino acids of the intestinal humoral pathways of satiety such as L-cell PYY secretion [6,18]. Nevertheless, as a protein-rich diet can often lead to varying levels of PYY secretion and satiety response, there may be also be co-regulatory factors interfering. Our present data, reveals an ability of *E. coli* ClpB to dose-dependently stimulate the PYY secretion, suggesting that this bacterial protein could be one of these regulatory factors. A factor that would depend on the composition of the gut microbiota i.e., on the presence of *E. coli* and possibly other species of ClpB-synthetizing bacteria. The biological significance of ClpB in the activation of host satiety (reducing nutrient intake) can be explained as a mechanism aimed at maintaining bacterial protein homeostasis. Of interest, our recent data showed that the *E. coli* ClpB protein, including the α-MSH-like motif, is specific to the order of *Enterobacteriales* and that the *Enterobacteriales* ClpB gene is depleted in gut microbiota of obese humans (submitted).

The phenomenon of the macronutrient effect on gut microbiota composition has been described as affecting preferential growth of certain bacterial families [19,20]. Similar with the macronutrient composition used in this study, it has been shown that supplementation of oleic acid increased the abundance of intestinal *Bifidobacteria* [21]. Moreover, mice fed with a diet rich in D-fructose have a lower proportion of Bacteroidetes compared to Proteobacteria [22]. A high-protein, low-carbohydrate diet resulted in increased rectal abundance of branched-chain fatty acids, reduced butyrate, and decreased *Roseburia*/*Eubacterium* [23]. Another rodent study also showed that rats fed protein-based foods isolated from cheese showed a significant increase in the numbers of *Bifidobacteria* or Lactobacilli in their feces [24]. No studies specifically looked at the impact of macronutrients on the *Enterobacteriales* proportion in the gut. In light of the present data, it would be interesting to see if *Enterobacteriaceae* could contribute to the beneficial anti-obesity effects of a protein-rich diet. 

With regard to the potential mechanism of action of *E. coli* ClpB in meal-induced activation of PYY secretion, it is likely that it involves a cephalic unconditional reflex [25]. In fact, considering the main location of the PYY secreting L-cells in the distal gut and their rapid activation after a meal (about 20 min), it is likely that these factors prevent their direct activation by the ingested nutrients. Thus, the cephalic reflex to ingestion by excretion of water together with previously digested nutrients and electrolytes in the gut lumen can be also involved in triggering the bacterial growth in the gut. According to such dynamics of bacterial growth, increased production of PYY, starting 20 min after nutrient provision, can be continuously stimulated by ClpB synthetized during the *E. coli* stationary phase. Indeed, our study showed that ClpB was increased in the bacterial supernatants after protein supplementation, supporting the idea that in vivo, ClpB can be released from both naturally lysed and alive bacteria in sufficient quantities to activate the enteroendocrine cells. Although the receptor linking the ClpB with PYY secretion has not yet been identified, it was shown that intestinal epithelial and L-cells express melanocortin receptors and that α-MSH can stimulate PYY secretion [26]. The presence of ClpB in the plasma of healthy humans further supports its physiological role in regulation of appetite [27]. Moreover, a slightly increased plasma ClpB level in patients with eating disorders correlated inversely with their eating disorder psychopathology and with anxiety scores supporting a direct physiological effect of this natural bacterial protein [27].

We also admit that the functional interpretations of the study results are limited to the use of a laboratory strain *E. coli* K12 bacteria and their derived ClpB protein. In fact, although *E. coli* K12 can be used as a model organism of human commensal Enterobacteria, it remains to be shown that macronutrients may influence ClpB production by those species of commensal *Enterobacteriaceae* which are commonly present in human gut and that their derived ClpB may activate the satiety pathways of the host. Moreover, considering the anaerobic conditions in the gut, we admit that the present in vitro model might not fully reflect the in vivo effects of macronutrients on ClpB production and the ClpB-mediated PYY secretion.

## 5. Conclusions

In conclusion, we have demonstrated that the macronutrient-dependent production of ClpB by *E. coli* was mainly due to the stimulatory effects of proteins on both ClpB mRNA and protein levels. There were no significant effects on ClpB production by fat or carbohydrates. Taken together with the dose-dependent effects of ClpB on PYY secretion from intestinal mucosal cells, our data support a possible link between *E. coli* ClpB and protein-induced satiety signaling in the gut.

## Figures and Tables

**Figure 1 nutrients-11-02115-f001:**
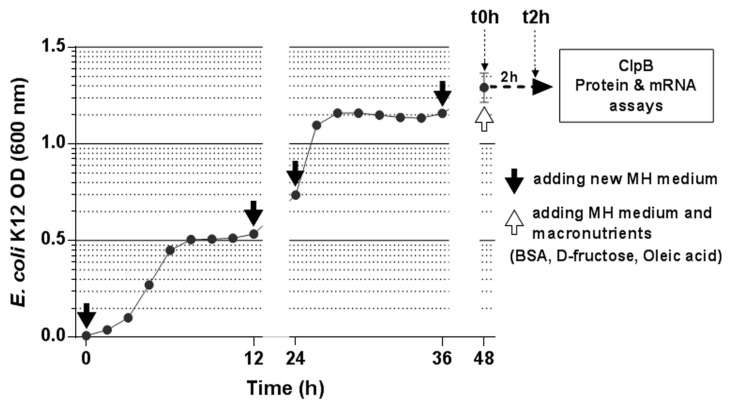
Growth curve of *E. coli* in continuous culture. *E. coli* K12 bacteria were cultured at 37 °C in Mueller-Hinton (MH) medium. Every 12 h, bacteria were supplemented with a new MH medium to imitate 2 daily meals (black arrow). After 48 h of culture (white arrow), ClpB mRNA and protein levels were analyzed immediately before (t0h) and 2 h after (t2h) supplementation of *E. coli* with MH medium of with isocaloric amounts of three macronutrients: BSA, D-fructose and oleic acid.

**Figure 2 nutrients-11-02115-f002:**
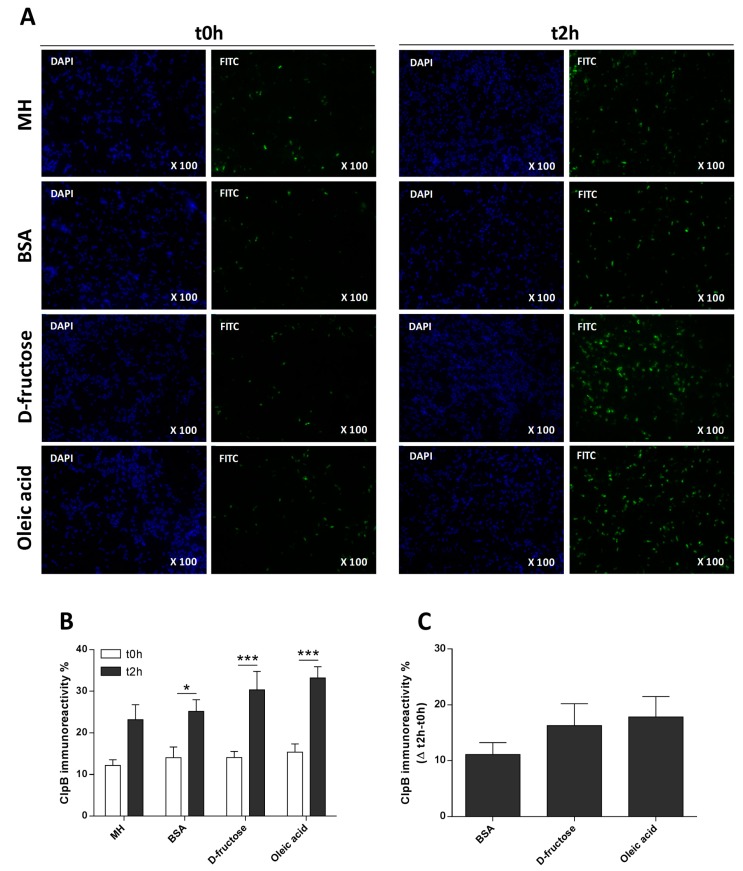
The effects of macronutrients on the ClpB immunoreactivity in *E. coli*. (**A**) Immunocytochemical detection of ClpB (FITC—green) in *E. coli* cells counterstained with DAPI (blue) at t0h (two left columns) and at t2h after supplementation with MH medium or macronutrients (two right columns). (**B**) The percentage of ClpB presence by the number of ClpB positive bacteria the total number of DAPI-stained bacteria (100%). (**C**) Relative to t0h increase in percentage of ClpB immunoreactivity at t2h after macronutrient supplementation. (**B**) *** *p* < 0.001, * *p* < 0.05; two-way ANOVA, Sidak’s multiple comparisons tests. MH t0h vs. t2h, unpaired *t*-test *p* < 0.05.

**Figure 3 nutrients-11-02115-f003:**
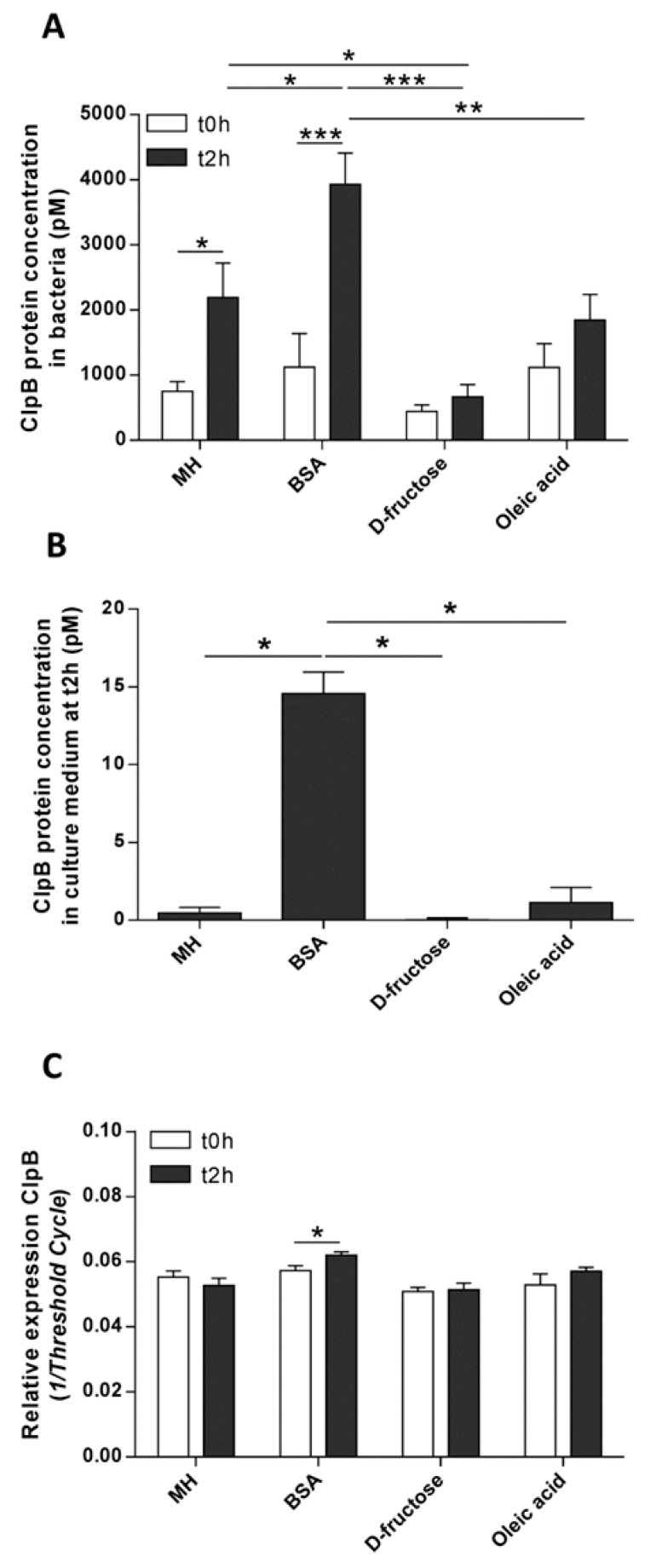
The effects of macronutrients on ClpB protein and mRNA expression. Concentrations of ClpB protein in bacteria (**A**) and in culture supernatants (**B**) before (t0h) and 2 h (t2h) after *E. coli* supplementation with MH medium and macronutrients. (**C**) The relative, between groups, levels of ClpB mRNA expression were estimated by the inverse values of the amplification cycle threshold (Ct) for each cDNA sample at t0h and at t2h. (**A**) * *p* < 0.05, ** *p* < 0.01, *** *p* < 0.001; two-way ANOVA, Sidak’s multiple comparisons test. (**B**,**C**) * *p* < 0.05, Mann–Whitney tests.

**Figure 4 nutrients-11-02115-f004:**
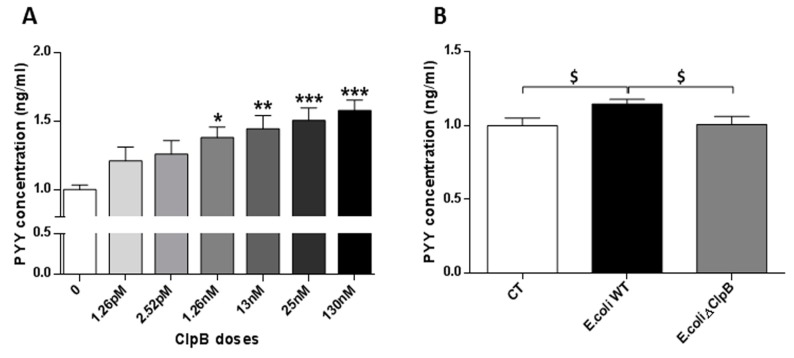
Effects of ClpB protein or *E. coli* total protein on peptide YY (PYY) secretion. PYY concentrations (ng/µL) in intestinal mucosal cell lysates after incubation with (**A**) increasing doses of *E. coli* recombinant ClpB protein or (**B**) with 15 ng/mL of total protein extracted from *E. coli* K12 wild type (WT) or *E. coli* K12 ΔClpB. For control condition (CT), cells were incubated with PBS. (**A**) * *p* < 0.05, ** *p* < 0.01, *** *p* < 0.001 (vs. basal concentration); one-way ANOVA, Sidak’s multiple comparisons test. (**B**) ^$^
*p* < 0.10, Mann–Whitney test.

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
