# Peer review of "Effects of Macronutrients on the In Vitro Production of ClpB, a Bacterial Mimetic Protein of α-MSH and Its Possible Role in Satiety Signaling"

_nutrients, 2019, doi:10.3390/nu11092115_

Round 1

Reviewer 1 Report

The authors investigated the effects of macronutrients on the in vitro production of C1pB and the possible role of C1pB in satiety signaling in a well designed study. The results show that protein, and not carbohydrates or fat, can increase C1pB mRNA and protein. Moreover, recombinant C1pB was able to increase PYY in intestinal cell cultures. The study contributes to the current knowledge and provides a new insight in the working mechanisms of possible weight management strategies. Additional clarity surrounding the choice of depiction of results, limitations of the study, and advice for further research could improve the manuscript. Moreover, readability could be improved by a few English language and style improvements.

Introduction

Line 59: The authors state that the fact that E. Coli growth followed by a stationary phase 2 hours after nutrient provision indicates that it is related to activation of the intestinal satiety pathways. Can they elaborate more on this hypothesis? How does a stationary phase of bacterial growth show that these bacteria play a role in this signaling? Is data missing here showing the satiety signaling after macronutrient infusion in the human gut (these data are available)? This might increase the plausibility of the link between the dynamics of bacterial growth and satiety signaling.

Results

Line 184: Data on the stationary growth phase 20 min after the 5th MH provision is not shown. However, in their discussion (line 251) the authors state finding the results of this dynamic growth after a meal as a validation of their previous found research. The authors should either remove this conclusion, because it is not supported by the data that is shown or the authors should include this data in the manuscript (a supplementary figure could suffice).

Line 189: The authors say mRNA and protein level analyses are depicted with a red arrow, however no red arrows can be found in figure 1.

Figure 2: why is MH not shown in figure 2C?

Line 206: “*p<0.05 (MH t0h vs t2h); Unpaired t-test (not shown)”. The depiction using * is the same as for the difference after BSA in line 205. However, data is not shown. Why the *? Why is this data not shown and why was an unpaired T-test used and was two-way ANOVA used for the other conditions?

Line 212 and line 215: Readability would be improved if the difference between the C1bP concentration and the C1pB protein concentration would be more clearly described. Is the concentration in line 212 the concentration in the bacteria and the concentration in line 215 the concentration in the medium?

Discussion

The discussion is mostly supported by the data shown. I would like to invite the authors to give insight in the limitations of the present study and to provide the readers with possible angles for further research.

Could the authors explain in the discussion why they think no effect was found when adding E. coli WT vs. e. coli ΔC1pB? Do the E. coli produce to little C1pB?

Line 251: As stated above, the authors here claim to have validated their previous experimental model. However, in my opinion this conclusion cannot be made based on data that is not shown. Please correct me if I am wrong.

Author Response

Reviewer 1

The authors investigated the effects of macronutrients on the in vitro production of ClpB and the possible role of C1pB in satiety signaling in a well-designed study. The results show that protein, and not carbohydrates or fat, can increase C1pB mRNA and protein. Moreover, recombinant C1pB was able to increase PYY in intestinal cell cultures. The study contributes to the current knowledge and provides a new insight in the working mechanisms of possible weight management strategies. Additional clarity surrounding the choice of depiction of results, limitations of the study, and advice for further research could improve the manuscript. Moreover, readability could be improved by a few English language and style improvements.

Response: We thank the reviewer for the comments and provide below point-by-point clarifications. The manuscript was also checked for readability by a native English speaker.

Introduction

Line 59: The authors state that the fact that E. Coli growth followed by a stationary phase 2 hours after nutrient provision indicates that it is related to activation of the intestinal satiety pathways. Can they elaborate more on this hypothesis? How does a stationary phase of bacterial growth show that these bacteria play a role in this signaling? Is data missing here showing the satiety signaling after macronutrient infusion in the human gut (these data are available)? This might increase the plausibility of the link between the dynamics of bacterial growth and satiety signaling.

Response: To further elaborate on this hypothesis we have added the following sentence from the existing data: “infusion of coli proteins 2h after onset of the stationary phase stimulated PYY secretion”. Unfortunately, no such data are available in humans.

Results

Line 184: Data on the stationary growth phase 20 min after the 5th MH provision is not shown. However, in their discussion (line 251) the authors state finding the results of this dynamic growth after a meal as a validation of their previous found research. The authors should either remove this conclusion, because it is not supported by the data that is shown or the authors should include this data in the manuscript (a supplementary figure could suffice).

Response: We are sorry for this inconsistency. The sentence in the discussion has been deleted and the corresponding text in the result section modified accordingly.

Line 189: The authors say mRNA and protein level analyses are depicted with a red arrow, however no red arrows can be found in figure 1.

Response: Sorry for this mistake coming from the draft version of the Figure 1. It was corrected and the “red arrow” was replaced by “white arrow” as shown in Figure 1.

Figure 2: why is MH not shown in figure 2C?

Response: We did not show MH because we wanted to illustrate the effects of only 3 macronutrients which was the study objective.

Line 206: “*p<0.05 (MH t0h vs t2h); Unpaired t-test (not shown)”. The depiction using * is the same as for the difference after BSA in line 205. However, data is not shown. Why the *? Why is this data not shown and why was an unpaired T-test used and was two-way ANOVA used for the other conditions?

Response: It was a mistake in the legend which was corrected. In fact the data are shown but the level of significance is indicated only in the legend of the Fig 2B.

Line 212 and line 215: Readability would be improved if the difference between the C1bP concentration and the C1pB protein concentration would be more clearly described. Is the concentration in line 212 the concentration in the bacteria and the concentration in line 215 the concentration in the medium?

Response: This is correct and for clarity, we further specified when ClpB concentration was assayed in bacterial cells.

Discussion

The discussion is mostly supported by the data shown. I would like to invite the authors to give insight in the limitations of the present study and to provide the readers with possible angles for further research.

Response: The following sentence with the limitations and further angles was included at the end of discussion: “We also admit that the functional interpretations of the study results are limited to the use of a laboratory strain coli K12 bacteria and their derived ClpB protein. In fact, although E.coli K12 can be used as a model organism of human commensal Enterobacteria, it remains to be shown that macronutrients may influence ClpB production by those species of commensal Enterobacteriaceae which are commonly present in human gut and that their derived ClpB may activate the satiety pathways of the host.”

 Could the authors explain in the discussion why they think no effect was found when adding E. coli WT vs. e. coli ΔC1pB? Do the E. coli produce to little C1pB?

Response: The effect of proteins extracted from E.coli WT was not significant, but showed a tendency of an increased PYY secretion at p<0.1. Importantly, such tendency was not observed when using proteins from ClpB-deficient E.coli, highlighting a stimulatory role of ClpB protein. The reason of a statistically non-significant stimulatory effects is probably due to the relatively low concentration of ClpB in the total protein preparation.

Reviewer 2 Report

Overall: The study is of interest given the recent associations of various gut bacterial strains with several pathological conditions and is part of the previous work conducted by some authors establishing the effects of E.coli on satiation signals, including PYY.

Comments:

Satiety is not measured in this study, therefore, statements  in the abstract and conclusion such “These data support a functional role of E.coli ClpB in mediating protein-induced satiety signaling in the gut” is premature, and should be tempered. Likewise, this study does not support the statement in the conclusion: “It also suggests that combination of protein-rich diet and ClpB-producing probiotics can be useful for more efficient appetite and body weight control” and should be removed. The use of probiotics to control weight gain, or in other pathologies for that matter, in humans, is still controversial and the data thus far is inconclusive.

Please provide justification for the nutrients chosen. The intestine does not ordinarily see oleic acid.

Line 18-18 correct  - MSH.  Alpha symbol is missing throughout.

Line 34-36 please rephrase it. It is incorrect to compare weight gain to fatness. Obesity is characterized by the amount of body fat, not weight gain.

Line 43-44, please correct grammar. It should be “….epithelium that secrete a variety…”

Line 48. I am not aware of such a concept as “superior satiety”. Please replace it.          

Line 69: Surprisingly is misspelled.

Line 73. The authors state “to address these questions…”, however there are no questions posed above the paragraph, only stated findings. The authors should identify the gaps and lay out the rationale for conducting this study.

Like 92, replace “At all”

Line 144, replace 7-w old male with “Seven-week old male”

Line 145, week should be abbreviated with “wk” and not “w”

Figure 1. Remove “figure 1” from the graph. It is already in the Figure legend. The same for all the other graphs.

Figure 1 Legend. The figure is black and white; however it refers to “red arrow”

Line 266 rephrase: “….as an independent…”

Line 278, the authors mentioned “satiation and satiety”. These are two distinct components of feeding behavior—while throughout the paper they refer to “satiety”. Please be consistent and/or explain the use of the two as appropriate and in the context of the results and discussion.

Line 294: replace: “By taking an example..” with “similar with the macronutrient composition used in this study…”

Line 305 -307, please rephrase for clarity

Author Response

Overall: The study is of interest given the recent associations of various gut bacterial strains with several pathological conditions and is part of the previous work conducted by some authors establishing the effects of E.coli on satiation signals, including PYY.

Comments:

Satiety is not measured in this study, therefore, statements in the abstract and conclusion such “These data support a functional role of E.coli ClpB in mediating protein-induced satiety signaling in the gut” is premature, and should be tempered. Likewise, this study does not support the statement in the conclusion: “It also suggests that combination of protein-rich diet and ClpB-producing probiotics can be useful for more efficient appetite and body weight control” and should be removed. The use of probiotics to control weight gain, or in other pathologies for that matter, in humans, is still controversial and the data thus far is inconclusive.

Response: We thank the reviewer for the comments. The conclusion on the functional role of E.coli ClpB in satiety signaling has been tempered as follows: “These data support a possible link between E.coli ClpB and protein-induced satiety signaling in the gut” and included in both the abstract and conclusion. The sentence on the potential utility of combining ClpB-producing probiotics with protein-rich diet has been deleted from the conclusion. Indeed, this is a speculation which was not yet supported by the data.

Please provide justification for the nutrients chosen. The intestine does not ordinarily see oleic acid.

Response: We agree that naturally occurring oleic acid is present mainly in form of esters (triglycerides), however, the appetite-modulating effects of lipids are thought to be mainly due to the action of free fatty acids formed after hydrolysis of triglycerides. This justifies the direct use of oleic acid in our experiment. We have included a clarification in the method section.

Line 18-18 correct  - MSH.  Alpha symbol is missing throughout.

Response: We have checked for the correct appearance of the alpha symbol throughout the manuscript, its absence was probably due to the automatic reformatting.

Line 34-36 please rephrase it. It is incorrect to compare weight gain to fatness. Obesity is characterized by the amount of body fat, not weight gain.

Response: weight gain was replaced by fatness

Line 43-44, please correct grammar. It should be “….epithelium that secrete a variety…”

Response: corrected

Line 48. I am not aware of such a concept as “superior satiety”. Please replace it.         

Response: “superior” was replaced by “best effect”

Line 69: Surprisingly is misspelled.

Response: to improve the readability, this sentence was modified

Line 73. The authors state “to address these questions…”, however there are no questions posed above the paragraph, only stated findings. The authors should identify the gaps and lay out the rationale for conducting this study.

Response: We agree with this comment and have formulated the questions before the study objectives.

Line 92, replace “At all”

Response: corrected

Line 144, replace 7-w old male with “Seven-week old male”

Response: corrected

Line 145, week should be abbreviated with “wk” and not “w”

Response: corrected

Figure 1. Remove “figure 1” from the graph. It is already in the Figure legend. The same for all the other graphs.

Response: This was due to a typical way how we submit figures to avoid the confusion during reviewing. The journal generated an outline that duplicated this information. For the revision we have deleted figure numbering.

Figure 1 Legend. The figure is black and white; however it refers to “red arrow”

Response: Sorry for this mistake coming from the draft version of the Figure 1. It was corrected and the “red arrow” was replaced by “white arrow” as shown in Figure 1.

Line 266 rephrase: “….as an independent…”

Response: corrected

Line 278, the authors mentioned “satiation and satiety”. These are two distinct components of feeding behavior—while throughout the paper they refer to “satiety”. Please be consistent and/or explain the use of the two as appropriate and in the context of the results and discussion.

Response: to avoid confusion and for consistency we kept only the term “satiety”.

Line 294: replace: “By taking an example..” with “similar with the macronutrient composition used in this study…”

Response: corrected

Line 305 -307, please rephrase for clarity

Response: the entire manuscript was checked for clarity by a native English speaker.

Reviewer 3 Report

The current study explored satiety effect of macronutrients on caseinolytic protease B (ClpB) release from E coli and its impact on intestinal satiety signaling in vitro. The authors previously demonstrated that E Coli chaperone ClpB plays an important role in satiety signaling pathways in response to feeding via intragastric infusion. The in vitro data are clear and straightforward to understand each presentation well. However, overall connection to reach a conclusion is somewhat loose and not strongly supported by the presented data. The authors should address the following concerns properly to make these in vitro data functional in a whole gastrointestinal setting.

1)    In their previous in vivo study, colonic infusion of E. coli proteins from the exponential growth phase failed to increase plasma PYY level. Therefore, it is questionable that the positive effect of BSA on PYY release from primary intestinal cells in vitro would warrant physiological relevance of the suggested pathway in vivo.

2)    The increase of ClpB mRNA expression in BSA treatment group looks very minimal even though the change is significant. How does the small increment of mRNA level contribute to large induction of bacterial protein and protein in the supernatant?

3)    Pure E. Coli culture released around 15 pM of ClpB into media and 1.26nM of ClpB was required to induce a significant PYY secretion from the intestinal cells. Due to small E. Coli population (approx. 0.1%) in gut microbiota, it is difficult to appreciate any contribution of the observed ClpB increase from E. Coli by protein ingestion on intestinal PYY release in real animal settings.

4)    Was the bacterial culture in an aerobic or anaerobic condition? Isn’t it supposed to be anaerobic to mimic in vivo gut setting. The gene and protein expression would be different between the two conditions in response to macronutrient challenges.

5)    Is fig 2 necessary for the manuscript? It doesn’t support any suggested pathways.

Author Response

The current study explored satiety effect of macronutrients on caseinolytic protease B (ClpB) release from E coli and its impact on intestinal satiety signaling in vitro. The authors previously demonstrated that E Coli chaperone ClpB plays an important role in satiety signaling pathways in response to feeding via intragastric infusion. The in vitro data are clear and straightforward to understand each presentation well. However, overall connection to reach a conclusion is somewhat loose and not strongly supported by the presented data. The authors should address the following concerns properly to make these in vitro data functional in a whole gastrointestinal setting.

Response: We thank the reviewer for the appreciation of our study and useful comments. Below we provide responses to all the concerns.

1)    In their previous in vivo study, colonic infusion of E. coli proteins from the exponential growth phase failed to increase plasma PYY level. Therefore, it is questionable that the positive effect of BSA on PYY release from primary intestinal cells in vitro would warrant physiological relevance of the suggested pathway in vivo.

Response: In our view the present results do not contradict the previous data and can be relevant to the physiological setting. In fact, BSA-stimulatory effect on ClpB production should be present beyond the exponential growth phase of bacteria and be the most prominent during the stationary phase as we show here at t2h. Accordingly, in the previous in vivo study, total coli proteins taken 2h of the stationary phase stimulated PYY secretion after their colonic infusion.

2)    The increase of ClpB mRNA expression in BSA treatment group looks very minimal even though the change is significant. How does the small increment of mRNA level contribute to large induction of bacterial protein and protein in the supernatant?

Response: The data on relative ClpB mRNA expression levels are based on the comparison of the threshold cycle for the transcript amplification. The absolute amount of mRNA can be underestimated by this approach.

3)    Pure E. Coli culture released around 15 pM of ClpB into media and 1.26nM of ClpB was required to induce a significant PYY secretion from the intestinal cells. Due to small E. Coli population (approx. 0.1%) in gut microbiota, it is difficult to appreciate any contribution of the observed ClpB increase from E. Coli by protein ingestion on intestinal PYY release in real animal settings.

Response: Considering the continuous renewal of the bacterial population in the gut following natural cycles of bacterial growth, we believe that the main source of ClpB available for the interaction with the intestinal satiety pathways of the host is the naturally lysed bacteria during their stationary phase. Furthermore, because the sequence of the coli ClpB protein (including a-MSH-like motif) is conserved in the order of Enterobacteriales and the relative prevalence of these bacteria in healthy gut is about 1-3% (in some pathological conditions can reach up to 80%), ClpB from periodically lysed enterobacteria should be theoretically sufficient to play a functional role in PYY-mediated satiety. We clarified this point in the discussion.

4)    Was the bacterial culture in an aerobic or anaerobic condition? Isn’t it supposed to be anaerobic to mimic in vivo gut setting. The gene and protein expression would be different between the two conditions in response to macronutrient challenges.

Response: We completely agree with this comment. The present experiment was performed in aerobic conditions as long as coli is a facultative anaerobe. Nevertheless, using anaerobic conditions would be more appropriate for the gut setting but experimentally are more challenging. We have included this statement in the study limitation.

5)    Is fig 2 necessary for the manuscript? It doesn’t support any suggested pathways.

Response: We think that this figure and corresponding discussion provides the necessary background for linking the bacterial biology with the host systems regulating satiety. As such, bacterial mechanisms of protein disaggregation can be linked with protein-induced satiety.

Round 2

Reviewer 3 Report

Nothing to comment